# Efficacy and Safety of Ceftazidime–Avibactam Alone versus Ceftazidime–Avibactam Plus Fosfomycin for the Treatment of Hospital-Acquired Pneumonia and Ventilator-Associated Pneumonia: A Multicentric Retrospective Study from the SUSANA Cohort

**DOI:** 10.3390/antibiotics13070616

**Published:** 2024-07-02

**Authors:** Marco Fois, Andrea De Vito, Francesca Cherchi, Elena Ricci, Michela Pontolillo, Katia Falasca, Nicolò Corti, Agnese Comelli, Alessandra Bandera, Chiara Molteni, Stefania Piconi, Francesca Colucci, Paolo Maggi, Vincenzo Boscia, Aakash Fugooah, Sara Benedetti, Giuseppe Vittorio De Socio, Paolo Bonfanti, Giordano Madeddu

**Affiliations:** 1Unit of Infectious Diseases, Department of Medicine, Surgery and Pharmacy, University of Sassari, 07100 Sassari, Italy; m.fois63@studenti.uniss.it (M.F.); f.cherchi25@studenti.uniss.it (F.C.); 2Fondazione ASIA Onlus, 20090 Buccinasco, Italy; ed.ricci@libero.it; 3Clinic of Infectious Diseases, Department of Medicine and Science of Aging, G. D’Annunzio University, Chieti-Pescara, 66100 Chieti, Italyk.falasca@unich.it (K.F.); 4Infectious Disease Unit, Fondazione IRCCS San Gerardo dei Tintori, 20900 Monza, Italy; n.corti7@campus.unimib.it (N.C.); paolo.bonfanti@unimib.it (P.B.); 5School of Medicine and Surgery, University of Milano-Bicocca, 20900 Monza, Italy; 6Infectious Diseases Unit, Fondazione IRCCS Ca’ Granda Ospedale Maggiore Policlinico, 20122 Milan, Italy; agnese.comelli@policlinico.mi.it (A.C.); alessandra.bandera@policlinico.mi.it (A.B.); 7Unit of Infectious Diseases, “A. Manzoni” Hospital, 23900 Lecco, Italy; c.molteni@asst-lecco.it (C.M.); s.piconi@asst-lecco.it (S.P.); 8Infectious Diseases Unit, AORN Sant’Anna e San Sebastiano, 81100 Caserta, Italy; f.colucci16@studenti.unisa.it (F.C.); paolo.maggi@unicampania.it (P.M.); 9Unit of Infectious Diseases, Garibaldi Hospital, 95124 Catania, Italy; enzoboscia@libero.it (V.B.);; 10Unit of Infectious Diseases, Santa Maria Hospital, 06129 Perugia, Italy; sara.benedetti@ao-siena.toscana.it (S.B.); giuseppe.desocio@ospedale.perugia.it (G.V.D.S.)

**Keywords:** HAP, VAP, ceftazidime/avibactam, fosfomycin, MDRO, pneumonia, combination, prolonged infusion

## Abstract

Hospital-acquired pneumonia (HAP) and ventilation-associated pneumonia (VAP) are challenging clinical conditions due to the challenging tissue penetrability of the lung. This study aims to evaluate the potential role of fosfomycin (FOS) associated with ceftazidime/avibactam (CZA) in improving the outcome in this setting. We performed a retrospective study including people with HAP or VAP treated with CZA or CZA+FOS for at least 72 h. Clinical data were collected from the SUSANA study, a multicentric cohort to monitor the efficacy and safety of the newer antimicrobial agents. A total of 75 nosocomial pneumonia episodes were included in the analysis. Of these, 34 received CZA alone and 41 in combination with FOS (CZA+FOS). People treated with CZA alone were older, more frequently male, received a prolonged infusion more frequently, and were less frequently affected by carbapenem-resistant infections (*p* = 0.01, *p* = 0.06, *p* < 0.001, *p* = 0.03, respectively). No difference was found in terms of survival at 28 days from treatment start between CZA and CZA+FOS at the multivariate analysis (HR = 0.32; 95% CI = 0.07–1.39; *p* = 0.128), while prolonged infusion showed a lower mortality rate at 28 days (HR = 0.34; 95% CI = 0.14–0.96; *p* = 0.04). Regarding safety, three adverse events (one acute kidney failure, one multiorgan failure, and one urticaria) were reported. Our study found no significant association between combination therapy and mortality. Further investigations, with larger and more homogeneous samples, are needed to evaluate the role of combination therapy in this setting.

## 1. Introduction

Hospital-acquired pneumonia (HAP) is the second most common nosocomial infection and the leading cause of death from nosocomial infections [1]. The incidence of HAP is between 5 and 20 cases per 1000 hospital admissions, with higher rates in immunocompromised, surgical, and elderly patients [1]. In addition, the incidence is 50% in patients with trauma and brain injury, probably related to altered consciousness and consequent microaspiration at the time of trauma [2].

Approximately one-third of nosocomial pneumonia are acquired in intensive care units (ICU) [1]. In particular, HAP is observed in 9–27% of intubated patients, four times more than in nonintubated subjects, with an estimated increased risk six to twenty-one times [1].

Concerning the etiology of ventilator-associated pneumonia (VAP), there is variability in the most frequent pathogens when considering the time of onset and admission.The early-onset forms of VAP (onset less than five days from intubation) are typically sustained by “community-acquired” microorganisms such as methicillin-sensitive Staphylococcus aureus, Streptococcus pneumoniae, and Haemophilus influenzae in about 55% of cases [3]. In the late-onset forms (more than five days from intubation), Pseudomonas aeruginosa, Klebsiella pneumoniae, and methicillin-resistant Staphylococcus aureus (MRSA) are the most frequent microorganisms associated with HAP and late VAP, followed by other Enterobacterales and other nonfermenting Gram-negatives such as Acinetobacter baumannii, Burkholderia cepacia, and Stenotrophomonas maltophilia [4]. 

Managing HAP and VAP is inherently complex, and it becomes particularly challenging when the causative microorganism is resistant to commonly used agents. The European Centre for Disease Prevention and Control (ECDC) annual report on antimicrobial resistance among European isolates indicates a rising trend in resistant microorganisms over the years, with the highest percentage observed in southern and eastern European countries. This is especially notable for third-generation cephalosporin resistance in *E. coli* and carbapenem resistance in *K. pneumoniae*, *P. aeruginosa*, and *A. baumannii* [5]. However, the frequency of HAP and VAP due to multidrug-resistant organisms (MDROs) is strongly influenced by local epidemiology; it can be profoundly different even between hospitals in the same region. Under antimicrobial pressure, Gram-negative bacteria, such as Enterobacterales (e.g., *E. coli*, *K. pneumoniae*), can develop various resistance mechanisms, including producing numerous beta-lactamases. These enzymes can break down a range of beta-lactams from penicillin to carbapenems and even newer agents like cefiderocol. Nonfermenters like *P. aeruginosa* also exhibit additional resistance mechanisms, such as altered porin levels, enhanced efflux pumps, and mutations, in penicillin-binding proteins [6].

To overcome the emergence of MDROs infections, new molecules were developed to be active against microorganisms producing carbapenemases [7]. One of the first compounds approved for treating such infection is avibactam (AVI), a novel beta-lactamase inhibitor. Avibactam is the only beta-lactamase inhibitor commercially available capable of inhibiting OXA-48 enzymes, although it does not act against class-B metalloenzymes such as NDM, IMP, and VIM [7,8]. For these enzymes, cefiderocol remains the only available drug [9]. Avibactam is available in a fixed association with ceftazidime (2 g of ceftazidime and 0.5 g of avibactam) and is approved in the EU for treating HAP and VAP. It is administered at a standard dosage of 2 g/0.5 g every 8 h [10,11].

Some studies suggest that the pharmacokinetic profile of ceftazidime/avibactam (CZA) may not provide adequate lung penetration [12]. In one study involving healthy volunteers, the epithelial-lining-fluid/plasma (ELF/plasma) concentration ratio was around 30% for ceftazidime and avibactam [12]. However, no similar data are available in a case of documented pneumonia, where tissue inflammation might enhance drug penetration. Other molecules with similar activity against carbapenem-resistant organisms (CRO), like meropenem/vaborbactam, imipenem/cilastatin/relebactam, and cefiderocol showed superior ELF penetration [13,14,15,16].

For this reason, in the case of pneumonia treated with CZA, adding another molecule that achieves good ELF penetration and with a synergic effect with beta-lactam could potentially improve efficacy and, hypothetically, prevent the emergence of resistance.

In vitro studies have reported synergistic activities of fosfomycin (FOS) with various classes of antibiotics, particularly with beta-lactams, daptomycin, linezolid, aminoglycosides, and tetracyclines [13]. However, this effect varies between the different molecules, even within the same class, and depends on the specific microorganism being considered [13]. Some in vitro studies have assessed the synergism between CZA and FOS against strains of Enterobacterales and *P. aeruginosa*, finding synergistic effects ranging from 0% to 67% and 25% to 100%, respectively, with no antagonist effect for any strain [17,18,19,20,21,22,23]. Another advantage of FOS, especially in treating lower airway infections, is its small molecular size, which facilitates good lung penetration [14]. Although this has been well documented in vitro, clinical data on the efficacy of combining FOS with beta-lactams or other molecules are scarce.

Given this background, we aim to evaluate whether the addition of FOS can enhance the efficacy of CZA in cases of HAP/VAP by comparing two cohorts (CZA monotherapy and CZA plus FOS) in terms of mortality rates and safety, as well as to identify which variables may most significantly impact the outcomes.

## 2. Results

Overall, 75 nosocomial pneumonia episodes in the database met the inclusion criteria and were included in the analysis. Of these, 34 (45.3%) received CZA in monotherapy, and 41 (54.7%) patients received a combination of CZA plus FOS (Table 1). 

The two treatment groups were similar regarding the Charlson Comorbidity Index (CCI), admission to ICU, septic shock, and continuous veno-venous hemofiltration (CVVH). Those on CZA alone were older (68 IQR 59–79) years vs. 62 (IQR 54–72) years; *p* = 0.06) and more frequently males (94.1% vs. 70.7%; *p* = 0.01). 

Among those with detected isolates (63/75 subjects), 54 (85.7%) were from the broncho-aspirate specimen (BAS) or broncho-alveolar lavage (BAL). In contrast, nine (14.1%) patients had microbiological identification of Gram-negative only from blood samples (seven *K. pneumoniae*, one *E. coli,* and one *P. aeruginosa*). Monomicrobial infection was present in 60 (95.3%) cases (27 *P. aeruginosa*, 31 *K. pneumoniae*, 1 *K. aerogenes*, 1 *E. coli*), while polymicrobial infection was identified in only three (4.7%) cases (one *K. pneumoniae* + *P. aeruginosa* + *E. coli*, one *K. pneumoniae* + *P. aeruginosa,* one *P. aeruginosa + D. acidovorans*). No significant difference between groups was observed regarding the number of polymicrobial infections and *K. pneumoniae* and *P. aeruginosa* isolates. 

In addition, 43 (68.2%) had a Gram-negative infection by carbapenem-resistant bacteria. Regarding *Enterobacterales*, in 28, the resistance mechanism was confirmed via molecular gene testing, 19 were *Klebsiella pneumonia* carbapenemase (KPC) producers, 2 were Oxacillinase-48 (OXA-48) producers, and 6 harbored both mechanisms. It is important to note that carbapenem-resistant microorganisms were more frequently treated with CZA+FOS than with CZA alone (74.3% vs. 60.7%; *p* = 0.03). 

In 12 individuals, no microorganism was detected, either from low pulmonary specimens (BAS/BAL) or from blood; therefore, the treatment was empirically set based on nonclinical isolates or other unknown risk factors. Seven of those had no microorganism identification, whereas five had various isolates from nonpulmonary or blood specimens (e.g., urine, bile).

Crude data of association with 28 survivals of the principal variables are displayed in Table 2. 

At the univariate analysis of 28-day mortality (Table 3), combination treatment was not associated with a significant mortality risk reduction (HR 1.14 (95% CI 0.46–2.83) *p*-value 0.78), whereas CZA prolonged infusion was associated with a lower risk of death (HR 0.41 (95% CI 0.16–1.04) *p*-value = 0.06).

The multivariate model was run for CZA+FOS vs. CZA monotherapy, adjusting for age, ICU admission, septic shock, and CZA prolonged infusion. In this model, we observed a lower risk of 28-day mortality in people treated with CZA+FOS without reaching the statistical significance (HR 0.32 (95% CI 0.07–1.39) *p*-value = 0.128) (Table 3). Additionally, we ran a multivariate model for prolonged infusion vs. standard infusion adjusting for age, ICU, septic shock and CVVH, resulting in a statistically significant reduction in 28-day mortality risk (HR 0.34, (95% CI 0.14–0.96), *p*-value = 0.03) (Table 4).

The Kaplan–Meier curves show no difference in the probability of survival at 28 days for CZA monotherapy vs. a combination of CZA+FOS (log-rank *p-*value = 0.8121) (Figure 1). However, when the Kaplan–Meier curve regarding 28 days of survival is performed for prolonged infusion vs. standard infusion, astatistically significant difference is observed in favor of prolonged infusion (log-rank *p*-value = 0.0493) (Figure 2).

Regarding the safety profile, three adverse events were reported: one acute kidney failure, one multiorgan failure in the CZA monotherapy group, and one urticaria in the CZA plus fosfomycin group.

## 3. Discussion

Our study provides an important insight into the treatment of HAP and VAP, particularly in the context of MDROs. Our findings showed that the combination therapy of CZA plus FOS was not associated with a statistically significant reduction in mortality at 28 days compared to CZA monotherapy. However, there was a trend toward improved outcomes with combination therapy. These results align with the evolving understanding of antibiotic therapy in managing severe hospital infections, where broader-spectrum and combination therapies are often considered to counteract resistance patterns.

The lack of significant difference in mortality rates between the monotherapy and combination therapy groups could be attributed to several factors. Firstly, the inherent potency of CZA against a broad range of pathogens might limit the additional benefit of including FOS. However, the pharmacodynamic synergy between CZA and FOS, suggested by in vitro studies, might still play a role in clinical settings by preventing the emergence of resistance during therapy. 

Secondly, it is important to note that we found an adjusted hazard ratio of 0.32 with a 95% CI of 0.07–1.39, suggesting that our sample may lack the power to detect a statistically significant difference. Therefore, a larger sample might have revealed a significant effect.

Thirdly, the two groups were unbalanced regarding the type of infusion and carbapenem-resistant microorganisms. In particular, the rate of carbapenem-resistant microorganisms was unbalanced towards the CZA+FOS combination group; this is an important factor that could have affected the efficacy of the treatment. Additionally, data on septic shock at baseline were missing for some patients (11.8% for monotherapy and 19.5% for combination treatment). 

Lastly, more patients in the combination treatment group were admitted to the ICU with no statistical difference. This could reflect a more critical baseline condition among those patients. 

In our study, treatment efficacy was in line with others where CZA-based regimens were evaluated, showing an overall success rate of 74.7% without significant differences among CZA and CZA+FOS-treated patients. In a meta-analysis, Wilson et al. stated that the mean clinical success rate of CZA regimens was 73% (CI % 67.7–78.4) [24]; however, the analyzed studies did not specifically evaluate combination treatment with CZA, nor did they evaluate efficacy in the context of HAP or VAP. 

In a retrospective observational study, Meschiari et al. evaluated the efficacy of FOS in combination with various other antibiotics for the treatment of any infection caused by MDR Gram-negative microorganisms; the survival rate at 28 days in those receiving of the CZA combination regimens was reported to be around 75%, a finding in line with our results [25].

In a recent review, Aslan et al. described clinical studies that compared CZA monotherapy versus CZA combination for the treatment of any infection site. They found that 30-day mortality ranged between 6.3% and 47.6% for CZA monotherapy and between 0% and 44.0% for CZA combination regimens. Notably, most of those studies evaluated a combination regimen composed of one or more active in vitro antibiotics; consequently, only a few studies assessed a specific combination strategy [26]. In this regard, Oliva et al. evaluated the efficacy of CZA monotherapy and CZA+FOS in the settings of bloodstream infections caused by KPC producer *K. pneumoniae*; they found no difference in terms of mortality at 30 days from the onset of bloodstream infections (BSI) (18% and 14.8% for monotherapy and combination, respectively, *p* = 0.807) and clinical cure (60.7% and 75.4% *p* = 0.120). Regarding the lower mortality compared to other studies, they stated that it could be due to the higher prevalence of urinary tract infections in their cohort [27,28,29].

Several in vitro studies assessed the synergistic effect of CZA+FOS against strains of MDR Gram-negative organisms. Ojdana et al. assessed the synergism of CZA with tigecycline, ertapenem, and FOS against 19 carbapenemase producers *K. pneumoniae* strains using E-test MIC:MIC ratio synergy method and found that ertapenem and fosfomycin had the most synergistic effect. Fosfomycin was found to have a synergistic impact even for NDM producer’s strains [21]. Similarly, Mikhail et al. evaluated the synergism of CZA in combination with meropenem, colistin, amikacin, aztreonam, and FOS against MDR strains of *P. aeruginosa* and *K. pneumoniae* by time-kill assay. They found that FOS had a synergistic effect for many strains, but the effect was stronger against *K. pneumoniae* strains than for *P. aeruginosa* strains [19]. Finally, Winkler et al. found that fosfomycin restored in vitro susceptibility of CZA against CZA-resistant strains of *P. aeruginosa* [30].

Our study found an interesting correlation between survival at 28 days after treatment started and receiving CZA with prolonged infusion. Additionally, those in the combination group received significantly less of this kind of infusion, which could have contributed to unbalancing the positive effect of fosfomycin on survival. Unfortunately, we cannot tell with certainty why those with monotherapy practiced prolonged infusion more often. We can speculate that the choice could be likely influenced by the difference in routine practice among the participating centers.

Regarding the positive effect of prolonged infusion of CZA in critically ill patients, Gatti et al. found that in a case series of 10 patients with carbapenem-resistant infections (5 BSIs, 4 VAP, 1 BSI+VAP), the prolonged infusion of CZA managed to achieve an optimal PK/PD target for 80% of patients [31]. 

Other evidence of a clinical benefit for extended or continuous infusion of CZA came from retrospective observational studies, case series, and case reports [32]. Tumbarello et al., in a cohort of 577 patients treated with CZA alone or in combination with other antibiotics, found that infusion of >3 h of CZA was associated with a reduced risk of 30 days mortality compared with standard infusion [28]. Xu et al. conducted a retrospective study regarding severe hospital-acquired infections due to carbapenem-resistant or difficult-to-treat *P. aeruginosa* treated with CZA alone or in combination with other agents; similar to our results, they found no difference in terms of clinical cure between monotherapy and combination therapy, while they found a statistically significant benefit for those who received CZA loading dose (OR = 0.03; 95% CI = 0.004–0.19; *p* < 0.001) and for CZA administration by prolonged infusion (OR = 0.15; 95% CI = 0.03–0.77; *p* = 0.002) [33]. Goncette et al. described a case series of ten patients treated with continuous infusion of CZA for the treatment of MDROs infections and reported a clinical cure rate of 80% and microbiological eradication rate of 90%; interestingly, three patients received a continuous infusion in an outpatient setting and all of those achieved clinical cure [34]. Finally, P. Lodise, via hollow fiber infection model, studied the effect of various dosing strategies of CZA plus aztreonam against MDR *E. coli* and *K. pneumoniae* strains and found that infusion longer than 2 h achieved greater bacterial killing than 30 min infusion, with the best effect achieved via continuous infusion. They also found that continuous infusion could suppress the emergence of resistant clones for over seven days [35].

Regarding renal replacement, some clinical studies found that being on substitutive renal replacement during CZA treatment was a factor with an increased risk of treatment failure and mortality [28,36,37]. Those studies were in line with our findings; in fact, being on CVVH during CZA treatment was associated with an increased risk of mortality at 28 days from treatment start in the univariate model (HR = 2.74, CI = 0.99–7.62, *p* = 0.05).

Regarding the safety profile of the combination treatment, we reported only three adverse events in our study. Skin rash is an adverse effect reported in less than 5% for ceftazidime alone, while a direct effect on renal function was reported in less than 1 person in 10,000 [12]; therefore, it may be more likely associated with the gravity of the infection and the severity of the basal condition of the patients rather than to the treatment itself, and the same could be said for multiorgan failure, as it is not an event correlated with either antibiotic. 

While our findings did not show a statistically significant reduction in mortality at 28 days with the combination therapy of ceftazidime/avibactam (CZA) plus fosfomycin (FOS) compared to CZA monotherapy, there was a trend toward improved outcomes with the combination therapy. Current treatment guidelines from organizations such as the Infectious Diseases Society of America (IDSA) and the American Thoracic Society (ATS) recommend broad-spectrum antibiotics, including CZA, for treating suspected or confirmed MDR Gram-negative infections [2,38]. Adding FOS to CZA is not routinely recommended. Still, it may be considered in cases where there is concern for inadequate coverage or in settings with a high prevalence of difficult-to-treat pathogens. Our study suggests that while adding FOS did not significantly reduce mortality, the observed trend towards better outcomes indicates that combination therapy may still be beneficial in specific clinical scenarios, mainly where the risk of resistance is high.

Furthermore, our finding of a significant benefit with prolonged infusion of CZA supports the current guidelines that advocate for optimized dosing strategies to enhance the pharmacokinetic/pharmacodynamic profile of antibiotics in critically ill patients. In conclusion, our results highlight essential considerations for optimizing therapy in high-risk patients, including the potential role of combination therapies and prolonged infusion strategies. We believe that our findings provide valuable information that may inform future updates to clinical guidelines, emphasizing the importance of individualized treatment approaches in managing HAP and VAP caused by MDR Gram-negative bacteria.

Our study has some limitations: first, it is a retrospective observational study with all its inherited biases. Secondly, the choice of the antibiotic regimen was likely influenced by a broad spectrum of factors that cannot be measured, which probably were mainly related to the severity of the conditions of the patients or the local clinical practice of single centers, since the addition of an extra agent for the treatment of MDROs infections is not a codified practice. Another limitation regards the diagnosis of HAP or VAP that was not defined by standardized criteria, and that could have led to a misidentification of some of the diagnoses, both in terms of over- and underestimation. Another issue is that some data were missing, especially data on the patients' clinical condition and vital parameters, or data about susceptibility to CZA or FOS. Concerning that, not every laboratory had the resources to test those agents; the reference method for fosfomycin susceptibility testing is agar dilution, a time-consuming procedure that needs expert personnel to be performed, and, therefore, it cannot be applied routinely. Finally, some of the treatments were set empirically, probably based on patients’ risk factors, local prevalence of KPC or OXA-48 producers, or precedent nonclinical positivity (e.g., rectal swab screening or upper-airways colonization). 

Despite these limitations, our study highlights the relevance of the administration of prolonged infusion of ceftazidime avibactam for the treatment of infection sites where it is known that the penetration could be insufficient, especially in the context of altered distribution like in the case of septic shock or when the patient needs continuous renal replacement treatment. Theoretically, this concept could be extended to other molecules with time-dependent pharmacodynamic characteristics. Further clinical investigation is needed to evaluate the role of continuous or prolonged infusion for treating pneumonia or other challenging infections in terms of pharmacokinetics. Continuous infusion could also improve the efficacy when facing MDR microorganisms and reduce the emergence of new resistant clones. Continuous infusion can obtain a stable concentration of antibiotics over the microorganism MIC. In contrast, intermittent bolus, despite a higher initial concentration, tends to have a concentration below the MIC for an excessive amount of time [31,35].

## 4. Materials and Methods

We conducted a retrospective cohort study using data from SUSANA (Surveillance of Safety and Outcome of New Antibiotics), a multicentric cohort, focused on patients treated with new molecules active against MDROs. These treatments include meropenem/vaborbactam, cefiderocol, imipenem/relebactam, ceftazidime/avibactam, ceftolozane/tazobactam, dalbavancin, and fosfomycin. The purpose of SUSANA is to evaluate the safety and efficacy of novel antimicrobial agents by collecting data through an online database. To date, 17 Italian centers have participated in this project, which has collected data on more than 662 treatments.

We included all patients diagnosed with HAP or VAP who received treatment with CZA in monotherapy or combined with FOS for at least 72 h. The determination of whether patients had HAP or VAP was not guided by uniform diagnostic standards but was determined at the discretion of the clinicians who collected patient data.

Data collected included age, sex, Charlson Comorbidity Index (CCI), comorbidities (previous myocardial infarction, congestive heart failure, chronic obstructive pulmonary disease, peripheral vascular disease, dementia, complicated or uncomplicated diabetes mellitus, history of cerebrovascular accident, connective tissue disease, peptic ulcer disease, liver disease, hemiplegia, moderate or severe chronic kidney disease, solid tumor, leukemia, lymphoma, and AIDS), hospitalization in the precedent 12 months, previous major surgery in the past 90 days, use of antibiotics in the precedent 90 days, state of immunosuppression during the treatment, presence of invasive devices (central venous catheter, arterial catheter, vesical catheter, endotracheal intubation), ICU admission, use of continuous veno-venous hemofiltration during treatment (CVVH), microbial isolates and their antimicrobial susceptibility, dosage of CZA and FOS, infusion method of CZA (standard or prolonged), presence of septic shock at the start of treatment, adverse effect (intended as any event that the clinician suspected could be linked to the treatment), all-cause mortality, date of death, and date of discharge.

The primary endpoint was to evaluate the difference in all-cause mortality at 28 days from the start of treatment between patients receiving monotherapy with CZA and those receiving combination therapy with CZA plus FOS. Secondary endpoints were to (i) evaluate differences in 28-day all-cause mortality between people who received CZA as a prolonged or continuous infusion versus those who received it as a standard 30-min infusion; (ii) evaluate the safety of CZA alone compared to CZA combined with FOS. 

### 4.1. Ethical Committee

The original study protocol was approved on 24 January 2019, by the coordinating center Ethics Committee (Brianza EC), and after that by all participating centers. Informed consent was obtained from subjects involved in the study, unless the subjects were deceased at the time of data collection. The study was conducted in accordance with the ethical standards laid down in the 1964 Declaration of Helsinki and its later amendments and by Italian national laws. Amendments were approved on 10 December 2020 by Brianza CE, and on 17 April 2024 by the Local Ethics Committee Lombardia 3.

### 4.2. Statistical Analysis

Data are presented using mean and standard deviation (SD) for normally distributed continuous variables, median and interquartile range (IQR) for non-normally distributed continuous variables, and frequency (%) for categorical and ordinal variables. Differences between groups for continuous variables were analyzed using the analysis of variance or Mann–Whitney U-test, based on the normality of distribution. The chi-square test or Fisher's exact test was employed for categorical variables, as appropriate.

Survival at 14 and 28 days from the start of treatment was evaluated using Kaplan–Meier curves and the log-rank test. The effects of CZA alone and the modality of administration were analyzed through hazard ratios (HRs) with 95% confidence intervals (CIs), derived from separate standard Cox regression models adjusted for covariates for each defined endpoint. All p-values were two-sided, with a threshold of *p* < 0.05 for statistical significance. Statistical analyses were performed using SAS/STAT software (version 9.4; SAS Institute.Inc., Cary, NC, USA).

## 5. Conclusions

To our knowledge, this is the only study comparing the efficacy of a specific combination of CZA versus monotherapy in the context of nosocomial pneumonia. However, our study did not show a significant association between CZA+FOS combination therapy and mortality reduction when compared to monotherapy to treat nosocomial pneumonia. However, these results could be partly explained by the small sample size and by the clinicians’ attitude toward prescribing combination therapy in critically ill patients. However, we showed some interesting findings regarding the infusion strategies of CZA. These could be matters of further investigation to assess the role of prolonged/continuous infusion to other new and old agents in the context of difficult-to-treat infections.

In conclusion, further investigations with a more solid study design and larger and more homogeneous samples are needed to evaluate the role of combining ceftazidime/avibactam with fosfomycin in the setting of nosocomial pneumonia. 

## Figures and Tables

**Figure 1 antibiotics-13-00616-f001:**
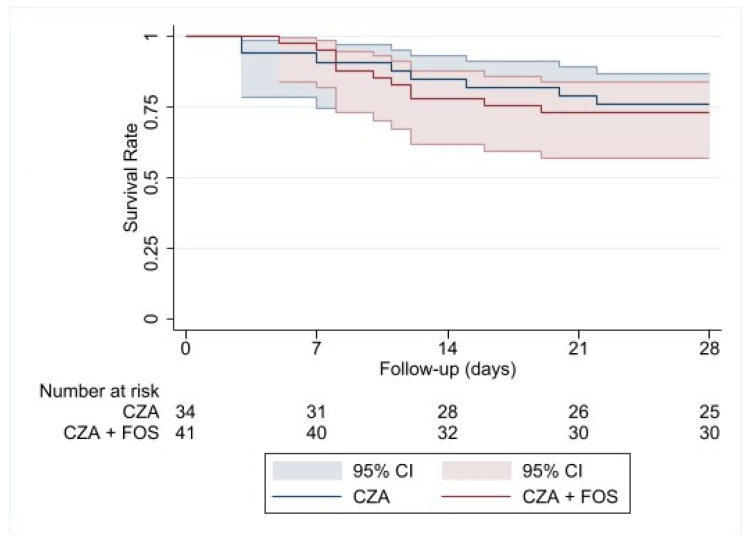
Kaplan–Meier curve on 28-day survival of ceftazidime/avibactam monotherapy vs. in combination with fosfomycin.

**Figure 2 antibiotics-13-00616-f002:**
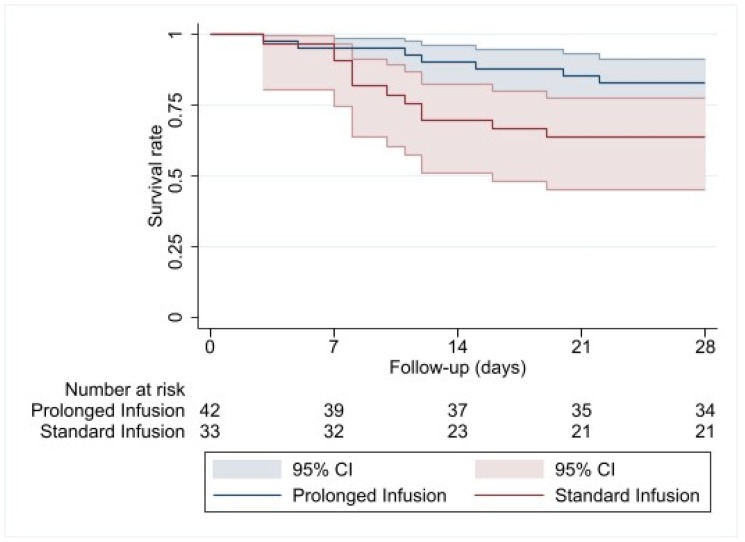
Kaplan–Meier curve on 28-day survival of standard infusion vs. prolonged infusion of ceftazidime/avibactam.

**Table 1 antibiotics-13-00616-t001:** Characteristics of 75 subjects included in the study.

Clinical Variables	Total N = 75	CZA N = 34 (45.3%)	CZA+FOS N = 41 (54.7%)	*p*-Value
**Males, n (%)**	61 (81.3)	32 (94.1)	29 (70.7)	0.01
**Age, years, Median (IQR)**	65 (57–73)	68 (59–79)	62 (54–72)	0.06
**CCI, Median (IQR)**	4 (2–6)	4 (3–6)	4 (2–5)	0.28
**ICU hospitalization, n (%)**	47 (62.7)	19 (55.9)	28 (68.3)	0.27
**Septic shock, n (%)** **Missing data, n (%)**	18 (24.0) 12 (16.0)	7 (20.6) 4 (11.8)	11 (26.8) 8 (19.5)	0.45
**Continuous venovenous hemodialysis, n (%)**	10 (13.3)	5 (14.7)	5 (12.2)	0.75
**Prolonged infusion, n (%)**	42 (56.0)	29 (85.3)	13 (31.7)	<0.0001
**Treatment duration, days, Median (IQR)**	12 (10–16)	11 (10–14)	13 (10–17)	0.24
**Microbiological features**				
**Patients with isolates from low pulmonary specimen (BAS/BAL), n (%)**	54 (72.4)	22 (64.7)	32 (78.0)	0.20
**Polymicrobial infections among patients with isolates (BAS/BAL), n (%)**	3 (5.6)	2 (9.1)	1 (3.1)	0.56
**Patients with isolates only from blood samples, n (%)**	9 (12.0)	6 (17.6)	3 (7.3)	0.17
**Patients with no microorganism detected *, n (%)**	12 (16.0)	6 (17.6)	6 (14.6)	0.72
**Patients with Gram-negative microorganism detected *, n (%)**	63 (84.0)	28 (82.4)	35 (85.4)	0.72
***Klebsiella* (31 *pneumoniae +* 1 *aerogenes*), n (%)**	32 (50.8)	13 (46.4)	19 (54.3)	0.26
***Pseudomonas aeruginosa* ^, n (%)**	28 (44.4)	12 (42.9)	16 (45.7)	
***Klebsiella pneumoniae + Pseudomonas aeruginosa* ^^ **	2 (3.2)	2 (7.1)	0	
***Escherichia coli*, n (%) **	1 (1.6)	1 (3.6)	0	
**Carbapenem resistant *, n (%)**	43 (57.2)	17 (60.7)	26 (74.3)	0.03
**-KPC producers **, n (%)**	19 (44.2)	8 (47.1)	11 (42.3)	0.66
**-OXA-48 like producers *, n (%) ***	2 (4.6)	0	2 (7.7)	
**-KPC + OXA-48 like producers **, n (%)**	6 (14.0)	2 (11.8)	4 (15.4)	
**-Other or unknown **, n (%)**	16 (37.2)	7 (41.2)	9 (34.6)	

CZA = ceftazidime/avibactam; FOS: fosfomycin; IQR: interquartile range; CCI: Charlson Comorbidity Index; ICU: intensive care unit; BAS = broncho-aspirate specimen; BAL = broncho-alveolar lavage; * Microorganisms detected from BAS, BAL, or blood samples. ** Percentages out of 43 carbapenem-resistant microorganisms. ^ +*Delftia acidovorans* in 1 subject; ^^ +*Escherichia coli* in 1 subject.

**Table 2 antibiotics-13-00616-t002:** Survival at 28 days after starting CZA/AVI treatment.

	Survival N (% of Total)	*p*-Value *
**Overall**	56 (74.7)	-
**Treatment** **CZA** **CZA+fosfomycin**	26 (76.5) 30 (73.2)	0.78
**Sex** **Male** **Female**	45 (73.8) 11 (78.6)	0.67
**ICU hospitalization** **No** **Yes**	27 (96.4) 29 (61.7)	0.01
**Septic shock** **No** **Yes Missing data**	35 (77.8) 12 (66.7) 9 (75.0)	0.35 0.82
**CVVH** **No** **Yes**	51 (78.5) 5 (50.0)	0.05
**Prolonged infusion** **No** **Yes**	21 (63.6) 35 (83.3)	0.06
**Carbapenem-resistant *** **No** **Yes** **Missing** **Not applicable (n = 12)**	11 (73.3) 32 (72.7) 5 (100) 9 (75.0)	0.81 n.e. 0.93
**Klebsiella pneumoniae **** **No** **Yes**	22 (75.9) 25 (73.5)	0.79
**Pseudomonas aeruginosa **** **No** **Yes**	24 (72.7) 23 (76.7)	0.69

* Cox proportional hazards model. ** out of 63 subjects with isolates. CZA = ceftazidime/avibactam; ICU = intensive care unit; CVVH = continuous veno-venous hemofiltration.

**Table 3 antibiotics-13-00616-t003:** Cox regression analysis estimates of CZA plus FOS with the risk of 28-day mortality, eventually adjusted for age, intensive care unit admission, septic shock, and prolonged infusion. HR: hazard ratio; CI: confidence interval; CZA = ceftazidime/avibactam; FOS: fosfomycin.

	Un-Adjusted			Adjusted		
	HR	95% CI	*p*-Value	HR	95% CI	*p*-Value
**CZA+fosfomycin,** **ref. CZA monotherapy**	1.14	0.46–2.83	**0.78**	0.32	0.07–1.39	**0.128**

**Table 4 antibiotics-13-00616-t004:** Cox regression analysis estimates of prolonged CZA infusion with the risk of 28-day mortality, eventually adjusted for age, ICU, septic shock and continuous veno-venous hemofiltration. HR: hazard ratio; CI: confidence interval; CZA = ceftazidime/avibactam.

	Un-Adjusted			Adjusted		
	HR	95% CI	*p*-Value	HR	95% CI	*p*-Value
**Prolonged infusion ** **ref. Standard infusion**	0.41	0.16–1.04	**0.06**	0.34	0.14–0.96	**0.04**

## Data Availability

The data that support the findings of this study are available from the corresponding author, upon reasonable request.

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
