# Peer review of "Efficacy and Safety of Ceftazidime–Avibactam Alone versus Ceftazidime–Avibactam Plus Fosfomycin for the Treatment of Hospital-Acquired Pneumonia and Ventilator-Associated Pneumonia: A Multicentric Retrospective Study from the SUSANA Cohort"

_antibiotics, 2024, doi:10.3390/antibiotics13070616_

Round 1

Reviewer 1 Report

Comments and Suggestions for Authors

The study carried out contributes to knowledge in the treatment of nosocomial pnemonia. All sections of the article are well conducted and written in accordance with scientific literature. I recommend that authors do not use acronyms in the title of the work.

Author Response

Reviewer (R) 1: The study carried out contributes to knowledge in the treatment of nosocomial pnemonia. All sections of the article are well conducted and written in accordance with scientific literature. I recommend that authors do not use acronyms in the title of the work.

Authors reply (AR): Dear reviewer, thank you for reading and reviewing our paper. We agree with your comment. The title of the paper is corrected as follows: “Efficacy and safety of ceftazidime-avibactam alone versus ceftazidime avibactam plus fosfomycin for the treatment of hospital-acquired pneumonia and ventilator-associated pneumonia: a multicentric retrospective study from the SUSANA cohort”

Reviewer 2 Report

Comments and Suggestions for Authors

Please perform the following corrections:

-            line 2: “in” doesn’t make sense

-            line 4: Abbreviations should not be used in the title, especially if they are not consecrated ones. The two types of pneumonia must be mentioned in full text, followed by abbreviations

-            line 27: HAP instead of hospital acquired pneumonia

-            line 28: VAP instead of ventilation-associated pneumonia

-            line 59: move “ventilator-associated pneumonia” to line 57, in front of “VAP”

-            line 98: is

-            line 122: “addition” instead of ”addiction”

-            line 132: delete the bracket in front of 68

-            line 136: please write in full what BAS means

-            line 137: please write in full what BAL means

-            line 146: Enterobacterales

-            line 147: please write in full what KPC and OXA means

-            Table 1: Patients with Gram-negative microorganism detected*, n (%) – total 63, 28 CZA, 36 CZA+FOS (total 64 – it is no concordant)

-            Table 1: aerogenes

-            line 163: please mention in the text a reference to table 2, because nothing is specified in the text about this table

-            line 163: CZA instead of CZA

-            Table 2: sex – male + female = 57 (Overall – 56!)

-            Table 2: In table 2, it is not understood how to calculate the percentages in the parenthesis. Please explain what all those percentages represent

-            line 170: CZA

-            line 238: CZA

-            line 239: CZA

-            line 241: CZA

-            line 245: CZA

-            line 255: . instead of , (14,8%)

-            line 255: please write in full what BSI means

-            line 311: “an event” instead of “and event”

-            line 402: CZA

-            line 407: CZA

Comments on the Quality of English Language

Very small mistakes, I mentioned them in the comments

Author Response

Please perform the following corrections:

-            OK line 2: “in” doesn’t make sense

-            OK line 4: Abbreviations should not be used in the title, especially if they are not consecrated ones. The two types of pneumonia must be mentioned in full text, followed by abbreviations

-            OK line 27: HAP instead of hospital acquired pneumonia

-            OK line 28: VAP instead of ventilation-associated pneumonia

-            OK line 59: move “ventilator-associated pneumonia” to line 57, in front of “VAP”

-            OK line 98: is

-            OK line 122: “addition” instead of ”addiction”

-            OK line 132: delete the bracket in front of 68

-            OK line 136: please write in full what BAS means

-            OK line 137: please write in full what BAL means

-            OK line 146: Enterobacterales

-            OK line 147: please write in full what KPC and OXA means

-Table 1: Patients with Gram-negative microorganism detected*, n (%) – total 63, 28 CZA, 36 CZA+FOS (total 64 – it is no concordant)

-            OK Table 1: aerogenes

-line 163: please mention in the text a reference to table 2, because nothing is specified in the text about this table

-            OK line 163: CZA instead of CZA

-            OK Table 2: sex – male + female = 57 (Overall – 56!)

-            OK Table 2: In table 2, it is not understood how to calculate the percentages in the parenthesis. Please explain what all those percentages represent

-            OK line 170: CZA

-            OK line 238: CZA

-            OK line 239: CZA

-            OK line 241: CZA

-            OK line 245: CZA

-            OK line 255: . instead of , (14,8%)

-            OK line 255: please write in full what BSI means

-            OK line 311: “an event” instead of “and event”

-            OK line 402: CZA

-            OK line 407: CZA

Authors reply: Dear reviewer. We would like to thank you for your carefull reading, and to provide your correction in order to improve the quality of our paper. We have deeply revised the paper, correcting it as you suggested.

Reviewer 3 Report

Comments and Suggestions for Authors

The authors have submitted a manuscript titled "Efficacy and safety of ceftazidime-avibactam alone versus in combination with fosfomycin for the treatment of hospital-acquired pneumonia (HAP) and ventilation-associated pneumonia (VAP): a multicentric retrospective study from the SUSANA cohort" for consideration to the journal Antibiotics. The study aims to evaluate the potential role of fosfomycin in combination with ceftazidime/avibactam in improving clinical outcomes in patients with HAP and VAP. Below are my comments

-         - The introduction is too long and not focused on the main aims and objectives of this study. Please attempt a more structured flow. The transition between the general background, pathogen-specific information, and treatment challenges needs to be more aligned. There is a long section of introductory material before FOS is mentioned. Consider starting  with the general background on HAP/VAP, move to pathogen-specific information, then discuss treatment challenges and resistance mechanisms, and finally, clearly state the study's objective and relevance.

-         -  While the introduction provides a solid background on the challenges of treating HAP and VAP with MDR pathogens, it would be beneficial to more explicitly discuss the in vitro findings on the synergistic effects of CZA and FOS. This could help strengthen the rationale for your research aims by clearly linking the preclinical evidence to the need for clinical evaluation. Specifically, mentioning previous studies that demonstrated synergistic effects in the introduction would provide a stronger foundation for investigating the potential benefits of combination therapy in your study. 

-          - When stating the aims and objectives, attempt to include all primary and secondary endpoints that will specifically be ascertained when comparing these cohorts. 

-         -  Please describe and add the mechanistic rationale behind adding FOS to CZA from a microbiologic/bacterial perspective besides achieving lung concentrations. See studies by Drs. Papp-Wallace/Bonomo. This may end up as a few lines in the discussion section or otherwise a shorter version in the introduction where you discuss the rationale behind selecting FOS.

-          - Given that this is a retrospective analysis, did you observe variations in the dosing of Fosfomycin in your center? There is no description clear to me on how Fosfomycin is administered and whether the utilized regimen is dosed appropriately to ensure optimal PK for the combination.

-           - Were there any cases in the combination cohort with missed doses of either CZA or FOS. This may inadvertently lead to suboptimal concentrations

-          - Does the study factor in susceptibility to Fosfomycin in the combination therapy protocol.

-          - Did this study analysis and multivariable model account for a change in therapy (i.e. de-escalation) when comparing both groups?

-          - Please clarify OXA-48 like 0% for the CZA cohort however, OXA-48 like +KPC had an n=2. This seems to be contradictory unless clarified.

-          - Can you clarify the underlying reasoning among these cases of using ceftazidime-avibactam in the treatment of infections that appear to be susceptible to carbapenems

-          - In the Cox regression analysis (Tables 3 and 4), could you explain the rationale for selecting the specific covariates used for adjustment? Were there any other potential confounders considered?

-          - The Kaplan-Meier curves (Figures 1 and 2) show survival probabilities. Could you provide more insight into the censoring of data and hhow it might have impacted the survival analysis?

-          - Were there any differences in the adherence to treatment protocols across the participating centers, and how were these managed in the analysis?

-          - In the discussion section, how might we interpret the clinical significance of these findings in the context of current treatment guidelines?

-          - Given that your study's overall success rate of 74.7% is in line with other CZA-based regimens and comparable to findings by Wilson et al. and Meschiari et al. Given that the studies referenced did not specifically evaluate the combination treatment with CZA and FOS for HAP/VAP, how are the similarities in success rates interpreted? Are there specific factors or study conditions that might explain these consistent results despite the diffferent treatment regimens and infection contexts?

-          -The discussion highlights the effect of prolonged infusion of CZA on patient survival. Considering the significant difference in the practice of prolonged infusion between the monotherapy and combination groups, please elaborate on how this difference might have influenced study results? If there is any data on the rationale behind the use of prolonged infusion in different centers, it should be mentioned in the manuscript including how might this have impacted the outcomes?

-         - In vitro studies have shown synergistic effects between CZA and FOS, against MDR GNR. This study did not find a statistically significant improvement in clinical outcomes with the combination therapy. please comment on any potential reasons for this difference

Comments on the Quality of English Language

The manuscript's English quality is generally good. Minor grammar corrections and punctuation would improve readability. 

Author Response

Reviewer (R) 3: The authors have submitted a manuscript titled "Efficacy and safety of ceftazidime-avibactam alone versus in combination with fosfomycin for the treatment of hospital-acquired pneumonia (HAP) and ventilation-associated pneumonia (VAP): a multicentric retrospective study from the SUSANA cohort" for consideration to the journal Antibiotics. The study aims to evaluate the potential role of fosfomycin in combination with ceftazidime/avibactam in improving clinical outcomes in patients with HAP and VAP. Below are my comments

Authors’ Reply (AR): Dear Reviewer, we would like to thank you for taking the time to read and review our paper. Your suggestions were fundamental in improving the quality of our work.

R:  The introduction is too long and not focused on the main aims and objectives of this study. Please attempt a more structured flow. The transition between the general background, pathogen-specific information, and treatment challenges needs to be more aligned. There is a long section of introductory material before FOS is mentioned. Consider starting  with the general background on HAP/VAP, move to pathogen-specific information, then discuss treatment challenges and resistance mechanisms, and finally, clearly state the study's objective and relevance.

AR: Thank you for your comment. We tried to follow your suggestions and provide a more concise introduction.

R: While the introduction provides a solid background on the challenges of treating HAP and VAP with MDR pathogens, it would be beneficial to more explicitly discuss the in vitro findings on the synergistic effects of CZA and FOS. This could help strengthen the rationale for your research aims by clearly linking the preclinical evidence to the need for clinical evaluation. Specifically, mentioning previous studies that demonstrated synergistic effects in the introduction would provide a stronger foundation for investigating the potential benefits of combination therapy in your study.

AR: We agree with your observation. While we have discussed the in vitro findings of this combination's synergism in the introduction, the mechanism can vary depending on the microorganism involved. Rather than analyzing all the different mechanisms, we believe it is important to emphasize the lack of clinical studies evaluating the advantage of this combination in specific situations compared to monotherapy

R: When stating the aims and objectives, attempt to include all primary and secondary endpoints that will specifically be ascertained when comparing these cohorts.

AR: We provided to modify the descriptions of aims and objectives in the last part of the introduction

R: Please describe and add the mechanistic rationale behind adding FOS to CZA from a microbiologic/bacterial perspective besides achieving lung concentrations. See studies by Drs. Papp-Wallace/Bonomo. This may end up as a few lines in the discussion section or otherwise a shorter version in the introduction where you discuss the rationale behind selecting FOS.

AR: thank you for your comment. The cited study by Papp-Wallace et al. provides evidence of the synergistic effects of fosfomycin and ceftazidime/avibactam, it does not go into detail about the specific mechanisms by which this synergism works. The authors primarily demonstrate the existence of synergism through in vitro and in vivo experiments, showing improved efficacy and reduced resistance when the drugs are used in combination. They hypothesize that the combination's effectiveness might be due to the combined targeting of different bacterial components and reduced mutation frequency, but they do not provide a detailed mechanistic explanation of how the synergism works at a molecular level. We added this part in the introduction

R: Given that this is a retrospective analysis, did you observe variations in the dosing of Fosfomycin in your center? There is no description clear to me on how Fosfomycin is administered and whether the utilized regimen is dosed appropriately to ensure optimal PK for the combination.

AR: thank you for your comment. There were no variations in the dosing of fosfomycin except in people with low eGFR or under CVVH; in those patients, it was adjusted according to the indication.

R: Were there any cases in the combination cohort with missed doses of either CZA or FOS. This may inadvertently lead to suboptimal concentrations

AR: thank you for your comment. We confirm that no dose in the two groups was missed.

R: Does the study factor in susceptibility to Fosfomycin in the combination therapy protocol.

AR: About susceptibility of fosfomycin, for most of the isolates there was no susceptibility testing, mostly because of the technical limitations of some laboratories. We considered this point as a limitation of our study in the discussion “Another issue is that some data were missing, especially data on the clinical condition and vital parameters of the patients, or data about susceptibility to CZA or FOS. Concerning that, not every laboratory had at the time the resources to test those agents; in fact, the reference method for fosfomycin susceptibility testing is agar dilution, a time-consuming method that needs expert personnel to be performed, and, therefore, it can’t be applied routinely.”

R: Did this study analysis and multivariable model account for a change in therapy (i.e. de-escalation) when comparing both groups?

AR: The dose changed in 4 people (5.3%), with 2 de-escalating (1 healing, 1 death) and 2 escalating (1 healing, 1 death). We did not includ this factor in the multivariable model.

R: Please clarify OXA-48 like 0% for the CZA cohort however, OXA-48 like +KPC had an n=2. This seems to be contradictory unless clarified.

AR: The categories were mutually exclusive. OXA-48 alone was not present in the CAZ group, were only 2 people with OXA-48+KPC were found. Overall, OXA-48 producers were 8 (6+2) in the overall cohort, 2 in the CAZ group (2+0) and 6 in the CAZ/FOS group (4+2).

R: Can you clarify the underlying reasoning among these cases of using ceftazidime-avibactam in the treatment of infections that appear to be susceptible to carbapenems

AR: Thank you for your comment. You’re right, some treatment was given as empiric treatment, in patient with rectal colonization with carbapenem resistant microorganism, or in patient with septic shock hospitalized in a time frame with a local cluster of carbapenem resistant microorganisms. Therefore, in some cases, the microorganism detected happens to be carbapenem-susceptible. The study, in fact, is not focused on the treatment of MDROs but on the efficacy in the context of nosocomial pneumonia.

R:  In the Cox regression analysis (Tables 3 and 4), could you explain the rationale for selecting the specific covariates used for adjustment? Were there any other potential confounders considered?

AR: Thank you for your comment. Age and the presence of septic shock were included as usual risk factors for survival, although they did not reach statistical significance in the univariate analysis, we still considered them important confounders. Additionally, hospitalization in the intensive care unit (ICU) was considered a confounder and was significantly associated with death in the univariate analysis. Regarding continuous infusion, we considered it a confounder for the efficacy of the two different approaches.

For the multivariate analysis of continuous infusion, we included age, septic shock, and ICU admission as confounders for the same reasons. Additionally, we included CVVH because we believe it could influence the choice between continuous and non-continuous infusion and potentially increase the risk of death.

R: The Kaplan-Meier curves (Figures 1 and 2) show survival probabilities. Could you provide more insight into the censoring of data and hhow it might have impacted the survival analysis?

AR: People who healed over the 28-day period or died after the considered period were attributed 28 days of survival. The only censored individual left the study because they had their empirical treatment interrupted after 5 days at the start.

R: Were there any differences in the adherence to treatment protocols across the participating centers, and how were these managed in the analysis?

AR: Due to the retrospective nature of the study, there was no standardized treatment protocol.

R: In the discussion section, how might we interpret the clinical significance of these findings in the context of current treatment guidelines?

AR: thank you for your comment. Our study provides important insights into the treatment of HAP/VAP. While our findings did not show a statistically significant reduction in mortality at 28 days with the combination therapy of ceftazidime/avibactam (CZA) plus fosfomycin (FOS) compared to CZA monotherapy, there was a trend toward improved outcomes with the combination therapy. Current treatment guidelines from organizations such as the Infectious Diseases Society of America (IDSA) and the American Thoracic Society (ATS) recommend broad-spectrum antibiotics, including CZA, for the treatment of suspected or confirmed MDR Gram-negative infections. The addition of FOS to CZA is not routinely recommended but may be considered in cases where there is concern for inadequate coverage or in settings with a high prevalence of difficult-to-treat pathogens. Our study suggests that while the addition of FOS did not significantly reduce mortality, the observed trend towards better outcomes indicates that combination therapy may still be beneficial in certain clinical scenarios, particularly where the risk of resistance is high. Furthermore, our finding of a significant benefit with prolonged infusion of CZA supports the current guidelines that advocate for optimized dosing strategies to enhance the pharmacokinetic/pharmacodynamic profile of antibiotics in critically ill patients. In conclusion, our results highlight important considerations for optimizing therapy in high-risk patients, including the potential role of combination therapies and prolonged infusion strategies. We believe that our findings provide valuable information that may inform future updates to clinical guidelines, emphasizing the importance of individualized treatment approaches in the management of HAP and VAP caused by MDR Gram-negative bacteria.
We have added this part to the discussion.

R: Given that your study's overall success rate of 74.7% is in line with other CZA-based regimens and comparable to findings by Wilson et al. and Meschiari et al. Given that the studies referenced did not specifically evaluate the combination treatment with CZA and FOS for HAP/VAP, how are the similarities in success rates interpreted? Are there specific factors or study conditions that might explain these consistent results despite the diffferent treatment regimens and infection contexts?

AR: Since in our study there was no statistical difference between the two groups, we felt confident when comparing the success rate with other studies where CZA-based treatments were analyzed. Furthermore Wilson et al. examined CZA based regimens regardless the presence of a companion agent, while Meschiari et al. examined fosfomycin based regimen (all combination regimens), therefore the latter could be rappresentative of the combination group, unfortunately there was no sub-analysis focused on the site of infection. These differences and similarities are further detailed in the discussion section of our paper.

R: The discussion highlights the effect of prolonged infusion of CZA on patient survival. Considering the significant difference in the practice of prolonged infusion between the monotherapy and combination groups, please elaborate on how this difference might have influenced study results? If there is any data on the rationale behind the use of prolonged infusion in different centers, it should be mentioned in the manuscript including how might this have impacted the outcomes?

AR: Thank you for your insightful comments. The significant difference in the practice of prolonged infusion between the monotherapy and combination groups likely influenced our study results. Prolonged infusion of CZA has been associated with improved pharmacokinetic/pharmacodynamic profiles and better patient outcomes, potentially explaining the lack of significant difference in mortality between the groups. There was no specific protocol for choosing between standard or prolonged infusion, and this variation likely reflects local clinical practices and the retrospective nature of our study. In our multivariate analysis, we included potential confounders such as age, ICU admission, septic shock, and continuous veno-venous hemofiltration (CVVH) to adjust for factors that could influence the choice of prolonged infusion and mortality.

R: In vitro studies have shown synergistic effects between CZA and FOS, against MDR GNR. This study did not find a statistically significant improvement in clinical outcomes with the combination therapy. please comment on any potential reasons for this difference

AR: Thank you for your comment. In the discussion, we proposed that the lack of statistically significant improvement with the combination therapy of CZA and FOS might be due to patients in the fosfomycin group having more critical baseline conditions, potentially influencing clinicians' choice of regimens. Although statistical significance was not reached, the adjusted odds ratio in our study favoured the fosfomycin combination. Additionally, other variables, such as specific resistance mechanisms varying between microorganisms, might have influenced fosfomycin's efficacy. Due to technical limitations, we do not have data on fosfomycin resistance for most isolates, as noted in the limitations section of the discussion.